# Research on the Durability and Reliability of Industrial Layered Coatings on Metal Substrate Due to Abrasive Wear

**DOI:** 10.3390/ma16051779

**Published:** 2023-02-21

**Authors:** Krzysztof Przystupa

**Affiliations:** Department of Automation, Lublin University of Technology, Nadbystrzycka 36, 20-618 Lublin, Poland; k.przystupa@pollub.pl

**Keywords:** layered organic coatings, resistance to abrasibility, reliability

## Abstract

This paper discusses the issue of evaluating the durability and reliability of organic coatings applied on the outer surfaces of roofing sheets. Two sheets, i.e., ZA200 and S220GD, were selected as research objects. Metal surfaces of these sheets are protected with multilayer organic coatings to protect them against weather conditions, assembly, and operational damages. The durability of these coatings was tested by evaluating their resistance to tribological wear using the ball-on-disc method. The testing was conducted in reversible gear according to a sinuous trajectory at a 3 Hz frequency. The test load was 5 N. When the coating was scratched, the metallic counter sample touched the metallic surface of the roofing sheet, which indicated a significant drop in electrical resistance. It is assumed that the number of performed cycles specifies the durability of the coating. Weibull analysis was applied to examine the findings. The reliability of the tested coatings was evaluated. The tests have confirmed that the structure of the coating is essential for the durability and reliability of products. The research and analysis included in this paper present important findings.

## 1. Introduction

The current stage of the development of societies faces the widespread use of industrial products, which is accompanied by the management of limited resources, resulting in social pressure to improve the value of the use of these industrial products [1]. Reliability and durability are requirements for all human-made products, equipment and structures [2]. If not satisfied, significant material losses and even risks to human health and life can occur [3]. Quality can be evaluated if product-specific evaluation measures are used. Evaluation measures can be physical quantities [4]. Reliability and durability are related to the resistance to the formation of defects and the duration of retention of functional characteristics and are important parameters in quality assessment [5]. Both of these parameters are related to the formation of damage due to use. The surfaces of metal components are prone to this type of damage. 

Metal surfaces show a limited resistance to environmental and climatic factors, so protective and aesthetic coatings are used to protect metal components. Outer coatings are exposed to loads due to use and environmental factors that can damage coatings. Abrasive wear can lead to damage to coatings, particularly over time of transportation, installation, and operation [6]. In some cases, if the coatings are exposed to windborne fine particles, erosive wear can also be a problem [7]. Modern coating systems usually consist of a priming coat, one or more intermediate coats, and a top coat [8]. Inorganic additives to the organic matrix are also known to improve properties, e.g., mechanical strength is improved, and the coefficient of thermal expansion is reduced [9]. Such coatings wear in complex ways that are difficult to predict. There is no single wear mechanism [10,11]. It is also known that coating hardness is not uniform, and it is considered stochastic in some works [12]; thus, analogically, wear is also considered stochastic [13]. Wear also depends on conditions in the tribological system and is generally not a material parameter [14,15]. In principle, it can be determined by comparative studies. Therefore, the durability and reliability of the coatings, especially multilayers and those composed of multiphase materials, are difficult to predict.

Bearing in mind the above, this paper is a comparative evaluation of multilayer industrial coatings applied on metal substrates. Both coatings are widely used in thin-walled structures of large and very large surface areas. The null hypothesis is that the structures of the coatings will be important for their durability and reliability, but the difference between the coatings will not be significant for both coatings and are intended for identical applications.

## 2. Material: Norms and Methodology

The research object is a new product that has not been examined by any other academic researchers. Industrial research findings are targeted differently, based on technical standards, and not intended for durability evaluation. Figure 1 shows how this paper’s research was managed.

The tested samples were taken from real industrial products. The material was coatings applied on sheet metal surfaces marked by their manufacturer, ArcelorMittal, as ZA200 (marked here as “black”—MAT 7024, steel and coating parameters are given in [16]) and S220GD (marked here as “white”—RAL 9010, steel and coating parameters are given in [17]). Figure 2 depicts how the organic coating layers are applied on the metal substrate.

The requirements for the product, its composition, properties, coating, quality, inspection, test methods, etc., are specified in the European standard, EN 10346:2015 [IDT] [18], and in the applied Polish national standard, EN 10346:2015-09 [19]. The products specified by this standard serve as the basis for organic coatings for flat products, regulated by EN 10169:2010 + A1:2012 [20] and PN-EN 10169:2022-08 [21] for construction and general technical applications.

There is also EN 10327:2004 [22], which specifies the requirements for continuous hot-coated products made of low-carbon steels for cold-forming, coated with zinc (Z), zinc–iron alloy (ZF), zinc–aluminum alloy (ZA), aluminum–zinc alloy (AZ), and aluminum-silicon alloy (AS) from 0.35 mm to 3.0 mm thick unless otherwise agreed. Thickness means the final thickness of the delivered product after coating.

Certain coating masses, according to EN 10346, are listed in Table 1.

Coating thicknesses can be calculated from the coating masses.

Resistance to cathodic corrosion of those coatings is directly proportional to coating thickness, i.e., steel is better protected against corrosion by a thick rather than fine coating. The weldability and deformability of products can be constrained by thinner coatings. Nevertheless, finer coatings are recommended in circumstances that require great ductility.

In the course of the study, the durability of the selected samples has been verified, estimating their resistance to tribology use. The abrasiveness of the coatings has been measured by a tribometer (Figure 3) with a ball-on-disc method. A kinematic pair consisted of a 6-millimeter ball of 100Cr6 bearing steel and a flat sample of a roofing sheet with a protective coating. The testing was conducted in reversible gear according to a sinuous trajectory at 3 Hz frequency. The test load was 5 N and was lower than the standard 10 N in the Taber test, considering that the tested coatings may be subject to low wear and damage simulation. The recording frequency of the quantity measurement was 30 Hz. The coating durability was determined by the number of friction cycles; however, a different approach to a number of friction cycles was taken than in alternative papers; for example, [13] indicated a fixed amount equal to 2000. In this paper, a resistance sensor was applied between the sample (coated roofing sheet) and the counter sample (ball). The coating material showed relatively high resistance (mainly polymeric organic coatings). When the coating was scratched, the metallic counter sample touched the metallic surface of the roofing sheet, which indicated a significant drop in electrical resistance. Up to this moment, the number of cycles was taken as coating durability.

## 3. Results and Discussion

### 3.1. Coating Structure

Figure 4 presents a cross-section of the S220 GD coating, while Figure 5 presents a cross-section of the ZA200 coating. Both of these coatings have a multi-layered structure. Inclusions from a few to a dozen micrometer measures are visible in the outer layer of the S220 GD coating. Oblong-shaped particles like fibers can be observed in its lower layer. It seems that the coating connection with the metallic substrate is good, and it is a type of adhesion in which the structure of the substrate has been modified, but not only in a mechanical way. As stated in [23], to increase substrate–coating adhesion, pro-adhesive layers or a befitting preparation of a substrate with mechanical and chemical methods can be used. It is very likely that in the case of the S220 GD coating, not only a mechanical preparation was applied. In composite coatings and materials, an interaction between a polymeric warp and reinforcement (fibers) is also significant [23], which can also be meaningful in the case of the S220 GD coating. The SEM image (Figure 4) shows the unstructured fibers, which, perhaps, had been torn out of a polymeric warp while grinding a metallographic specimen. This may point to their poor maintenance in the coating structure. In the abrasive wear tests, this kind of behavior of the reinforcement may influence the durability of the S220 GD coating.

The structure of the “black” (ZA200) coating is complex and layered (Figure 5). The outer layer coating has a multiphase structure, and it very likely includes organic and nonorganic stages. The particles representing the reinforcement-stage (bright fields) and the warp-stage (dark fields) particles are dispersed and sharply visible. The dispersion is even—agglomerates of particles are not observed [24]. An influence of those particles in wear tests under sliding friction conditions may be crucial. Their influence can be explained in the following manner: if the dimensions of filler particles and a space between them are smaller than deformations and distortions due to an interaction of two substances, then the material behaves in an inhomogeneous manner, and its resistance to wear is similar to that of the polymeric warp. If the filler particles, deformation scale or filler particles are approximately larger, the material behaves as heterogeneous, and the wear is less than the wear shown in the resin base [25]. 

A major impact of a filler in a polymeric warp and smaller particles was shown in its effects on the composite’s property surface improvements and provided good dispersion in the material. This can be confirmed in the case of the outer layer of the ZA200 coating. On the other hand, the large particles of a filler increase the coefficient of friction and, thereby, a frictional force [26]. The spaces between layers are clear, which means that layer adhesion is loose. Likewise, adhesion to metal substrates is discontinuous. It is possible that the structure of the coating layer directly adjoining the metallic substrate contains pores. Such a case may translate into the tribological durability and reliability of the coatings. 

### 3.2. Sliding Friction and Wear

Figure 6 and Figure 7 present the selected graphs of coating durabilities. The chart marked in a “black” line refers to the ‘ZA200’ sample, and the chart marked in a gray line refers to the ‘S220 GD’ sample. The drop in the electrical resistance to zero indicated total coating wear. Depending on sliding, the courses of variations in electrical resistance versus the friction cycles are noticeably different for both of the coatings.

Figure 8 and Figure 9 represent the frequency distribution bar charts of the sliding friction test results at a width specified in intervals depending on the material. Moreover, the bar charts present descriptive statistics, statistical measures of location (average value) and measures of variability (standard deviation as well as a minimal and maximum value).

The Grubbs test was used to quantify the outliers in the database [27] and verify if the test contained resulting values flawed by a gross error. It is based on hypothesis verification:-null—*H*_0_: no outliers in the database,-alternative—*H_a_*—outliers in the database.

A test statistic is as follows:$$T(X)=\frac{max|X_{i}-\overset{-}{X}|}{S_{x}}$$
where:

*X_i_*—research trial resulting value,

$\overset{-}{X}$—research trial average value,

*S_x_*—standard deviation.

The Grubbs test critical value is calculated from the formula:$$c_{\alpha }=\frac{N-1}{\sqrt{N}}\sqrt{\frac{t_{\frac{\alpha }{2N},N-2}^{2}}{N-2+t_{\frac{\alpha }{2N},N-2}^{2}}}$$
where: 

$t_{\frac{\alpha }{2N},N-2}^{2}$*—*1 − *α*/2*N* quantile of *t*-test distribution by *N* − 2 degrees of freedom,

*N*—number of measurements in a group.

Subsequently, a parameter with a higher value was compared with the Grubbs test critical parameter, which corresponds to a statistical sample size and selected probability level. If the experimental value is higher than the critical value, then a doubtful result is affected by the gross error and can be discarded [27]. The null hypothesis is discarded if:$$T\left(X\right)>c_{\alpha }$$

Table 2 presents the Grubbs test results (occurrence of outliers for *p* > 0.05) as well as the trimmed mean and Winsorized mean.

The *p* parameter had significantly higher values than the assumptive significance level (0.05). The Grubbs test results may point to outliers. This kind of resulting value can be verified by frame-type graphs with extreme and outlier observations. Furthermore, this type of graph allows for capturing information on the dispersion, positioning and symmetry distribution of the empirical variables in one figure. Figure 10 presents a frame-type graph of the roof sheets’ coating durability test results. Its analysis indicated the outliers, as marked with arrows. Two outliers were verified—one for each coating. It was considered that those values do not substantially affect the determination of which coating is more durable. Dispersion may arise from a local differentiation of the metal sheet coating’s roughness and thickness, as well as its mechanical properties. Analysis of the test results analysis indicated that the “black” coating had over twice the typical abrasion resistance. It should be pointed out that there was no difference in both of the coatings’ thicknesses, which was analyzed by microscopic examination with the results presented in Section 3.1.

The statistical Mann–Whitney (M-W) *U*-test evaluated the differences between the test results of coating durability. This test is the equivalent of the parametric *t*-test, which evaluates the differences between averages. This test verifies the hypothesis that the two analyzed samples are from different populations. It assumes that the analyzed variables can be ordered from the smallest to largest values, i.e., they are measured on an ordinal scale. Their interpretation is virtually identical to the *t*-test for independent samples [28]. 

The *U* statistics given in the table stand for the value of the M-W test applied to small numbers (<20), and Z denotes the value of the M-W test if the size of both groups is >20 [28]. Our analysis uses the *U* statistical parameter.

The *p*-value given in Table 3 is the significance level at which the null hypothesis of no difference between the results obtained in the groups can be rejected or accepted. The hypothesis can be rejected if *p* ≤ 0.05. 

The M-W test shows statistically significant differences between the results on the durability of the two coatings, which makes it possible to partially undermine the null hypothesis of the present study. The next step of the analysis of the test results was the Weibull analysis that followed a two-parameter Weibull distribution as a model for the probability distribution of non-destruction. The parameters were estimated by the maximum likelihood method [29]. The Weibull method is often applied because of its flexibility [30]. Figure 11 and Figure 12 show the non-parametric probability plots. The horizontal time-life axis (number of friction cycles) is logarithmically scaled, whereas the vertical axis shows log(log(100/(100 − F(t)))) (the left y-axis is described in terms of a probability scale given as a percentage). The parameters of the Weibull distribution are read from the graphs. The shape parameter equals the slope coefficient of the fitted straight line, and the scale parameter can be calculated as exp (absolute term/slope) [31].

The scale parameter (“Scale” in Figure 11 and Figure 12) expresses the characteristic value of coating durability, which corresponds to 63.2% of destruction cases [32]. The shape parameter—the Weibull modulus (“Shape” in Figure 11 and Figure 12) is taken as an indicator of the uniqueness (dispersion) of coating durability in our statistical sample [33]. A high value of the shape parameter indicates the likelihood of less variation in coating durability and, consequently, means potentially higher coating reliability. For both of the coatings, the value of the shape parameter of the Weibull distribution was higher than 1, which means that the tested coatings failed due to fatigue processes occurring in the friction process [34,35]. A higher value of the shape parameter (about 4) was obtained for the ZA200 coating, which means its higher reliability. 

A higher scale parameter was obtained for the same ZA200 coating, indicating its higher durability [36]. Figure 13 and Figure 14 show the course of the function of CPS, i.e., the cumulative proportion of surviving. The matching of the theoretical distributions was visually analyzed, and the linear, exponential, Weibull, and Gompertz models were compared—it was the latter model that matched best with both coatings [37,38]. Weights were also used, and the three colored lines in the graph illustrate the theoretical distributions that result from three different estimation procedures (least squares and two weighted least-squares methods). The Gompertz model can be used for composite structures subjected to cyclic loading [39].

The probability density (PD) was estimated in the next step of the analysis. This is the estimated probability of a defect in a given interval calculated per unit of time from the following formula:$$F_{i}=\frac{\left(P_{i}-P_{i+1}\right)}{h_{i}}$$
where:

*F_i_*—probability density in the *i*-th interval, 

*P_i_*—estimated cumulative proportion surviving at the beginning of the *i*-th interval (at the end of the *i* − 1 interval), 

*P_i+*1*_*—cumulative proportion surviving at the end of the *i* interval, 

*h_i_*—width of the interval.

Figure 15 and Figure 16 graphically show the results of the probability density calculations in ranges depending on the number of friction cycles. Matching to the weights was also carried out analogically, as it was in calculating the CPS function. The best visual matching of the Gompertz function was obtained from the PD analysis.

The next step in the durability analysis was to determine the hazard function. The hazard function is described by the probability per time unit that a case that has “survived” since the beginning of the study will fail in a given interval. The risk function, *h*(*t*)**, is called the intensity of the increase in unreliability *q*(*t*)** in relation to reliability *R*(*t*)**.
$$h(t)=\frac{dF/dt}{R(t)}$$
where:

*F*(*t*)**—unreliability,

*R*(*t*)**—reliability,

*t*—generalized lifetime (number of thermal cycles).

Figure 17 and Figure 18 show the hazard function.

It is clear from the compared Figure 17 and Figure 18 that the ZA200 (“black”) sample has a much better coating (more durable).

## 4. Summary

To sum up the conducted tests, it is necessary to relate the very unambiguous results to the quality of the product in use or simply to the durability of the roof, bearing in mind that:

Reliability is the probability that the objects will maintain the meaning of all the parameters that characterize the ability to perform the required functions under specified modes and conditions of use, maintenance, storage and transportation over a fixed period of time [40].

Reliability is also defined in terms of quality. Quality is a measure of variability up to its potential latent inconsistency or failure in a representative sample; thus, reliability is a measure of variability over time under operating conditions and is the quality that develops over time under operating conditions [41].

The reliability of products depends largely on how well the quality assurance measures have been planned, and reliability functions as a function of quality.

It is worth applying the QFD (quality function deployment) method, which is a technology for quality function deployment, quality function structuring, or product designing in which consumer demands are first identified, then followed by technical characteristics of products and production processes that best satisfy identified needs, which results in higher product quality [42]. The QFD method is useful for establishing and identifying the quality and reliability of products and coatings to be implemented in innovative designs.

This study was to benchmark the durability results of the selected technical coatings. The literature analysis and research result in the following final conclusions:The structure of the coating is fundamental for the durability and reliability of the product. The difference between the coatings is significant, it does not matter a good deal if both are intended for identical applications.The abrasion resistance of coatings likely depends on the layered structure of the coatings. The three-layer “black” coating showed higher abrasion resistance.Our analysis indicates that the ZA200 (“black”) sheet showed better durability and performance quality.

## Figures and Tables

**Figure 1 materials-16-01779-f001:**
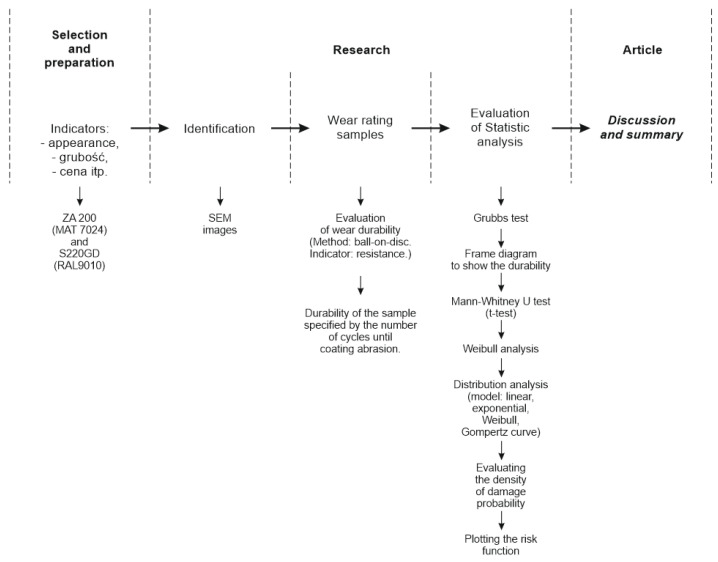
Schematic of the management of the research.

**Figure 2 materials-16-01779-f002:**
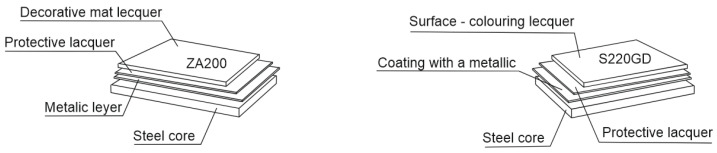
Schematic of applying the outer coatings on the sheet metal surfaces.

**Figure 3 materials-16-01779-f003:**
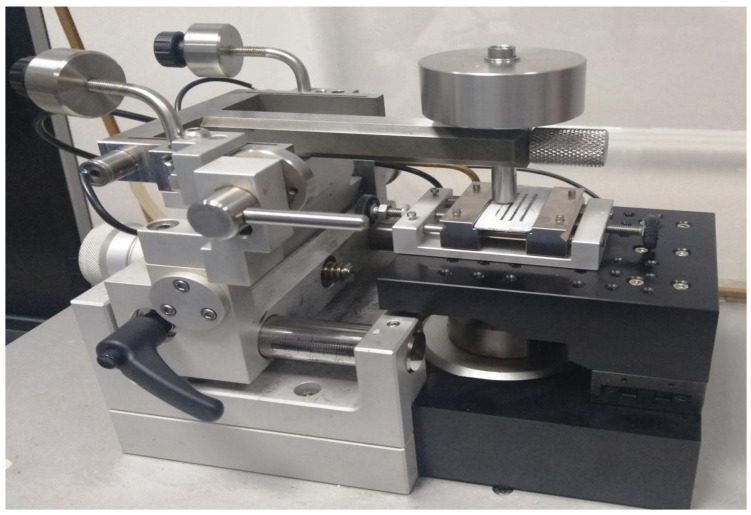
Tribometer used in the abrasibility testing of the coatings protecting roofing sheets.

**Figure 4 materials-16-01779-f004:**
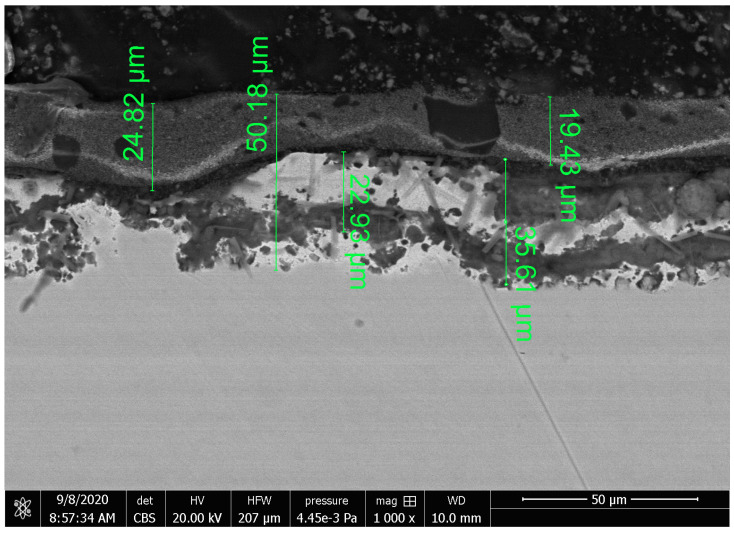
SEM image of the S220 GD coating cross-section (“white”).

**Figure 5 materials-16-01779-f005:**
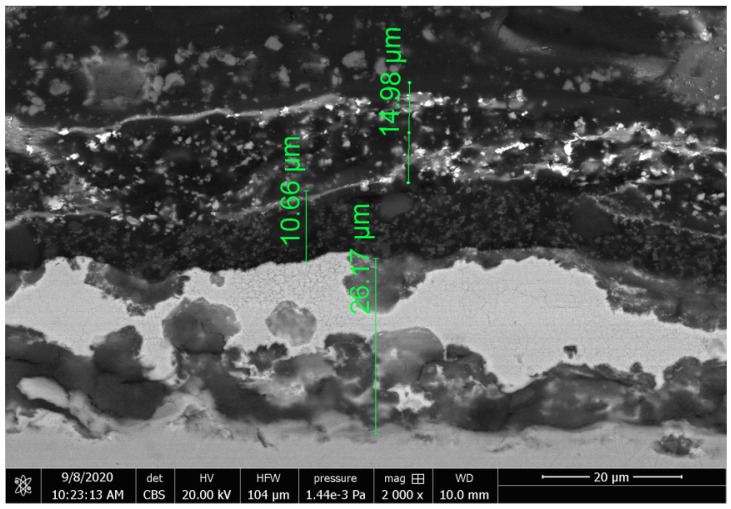
SEM image of the cross-section of the ZA200 coating (“black”).

**Figure 6 materials-16-01779-f006:**
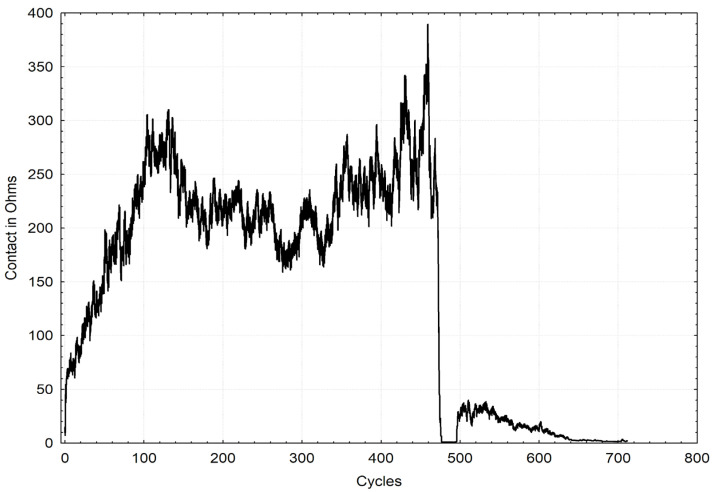
Selected graph of the durability of the “black” ZA200 coating.

**Figure 7 materials-16-01779-f007:**
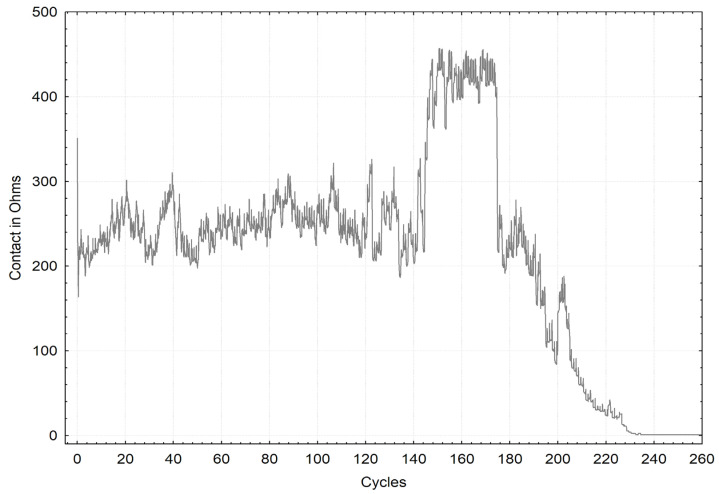
Selected graph of the durability of the “white” S220GD coating.

**Figure 8 materials-16-01779-f008:**
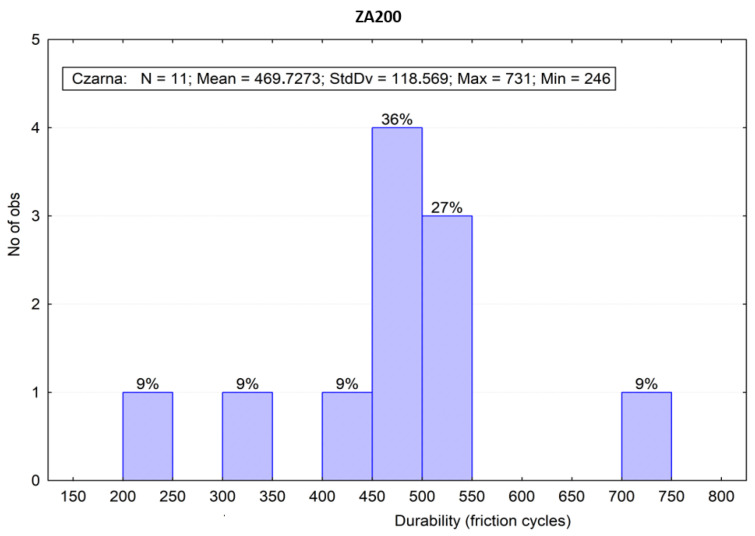
Frequencies of the sliding friction test results of the “black” (ZA200) coating.

**Figure 9 materials-16-01779-f009:**
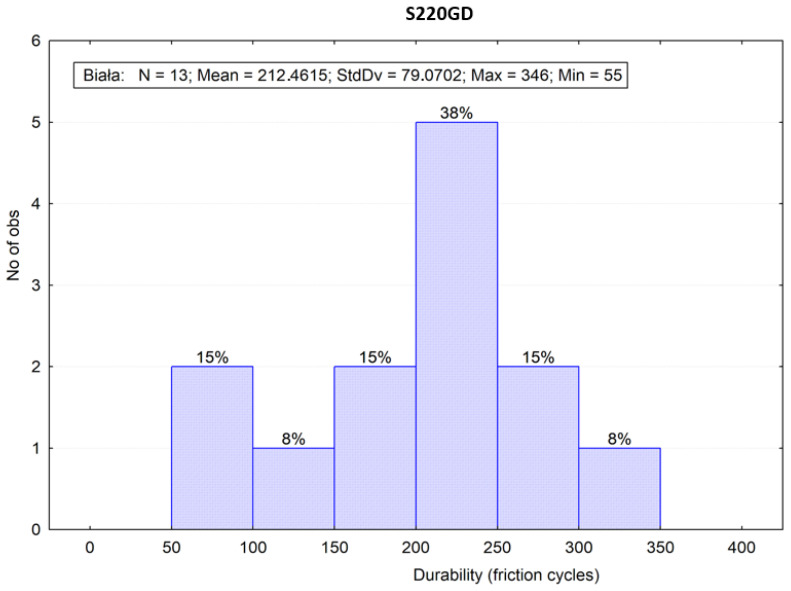
Frequencies of the sliding friction test results of the “white” (S220GD) coating.

**Figure 10 materials-16-01779-f010:**
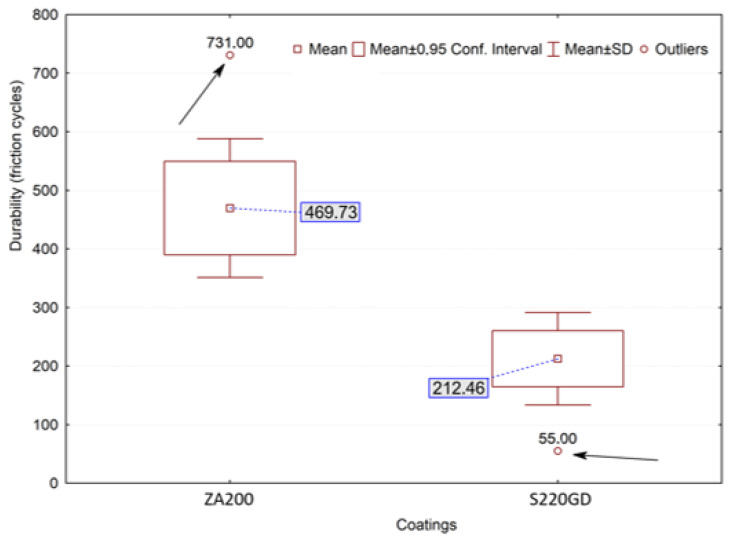
Frame graph of the results for the abrasibility of the sheet roofing coatings.

**Figure 11 materials-16-01779-f011:**
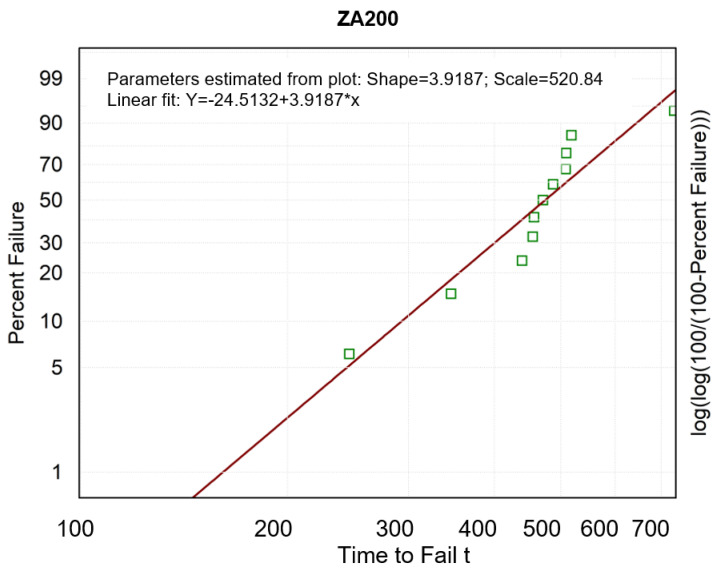
Weibull probability distribution for the wear of the ZA200 coating.

**Figure 12 materials-16-01779-f012:**
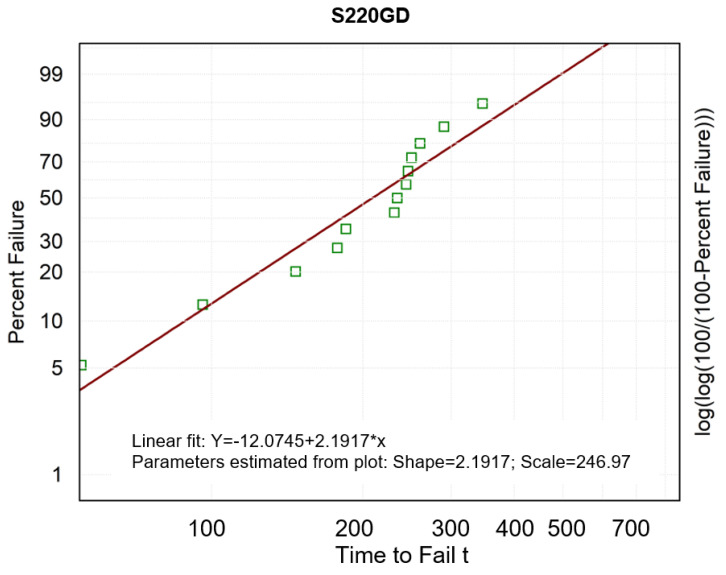
Weibull probability distribution for the wear of the S220 GD coating.

**Figure 13 materials-16-01779-f013:**
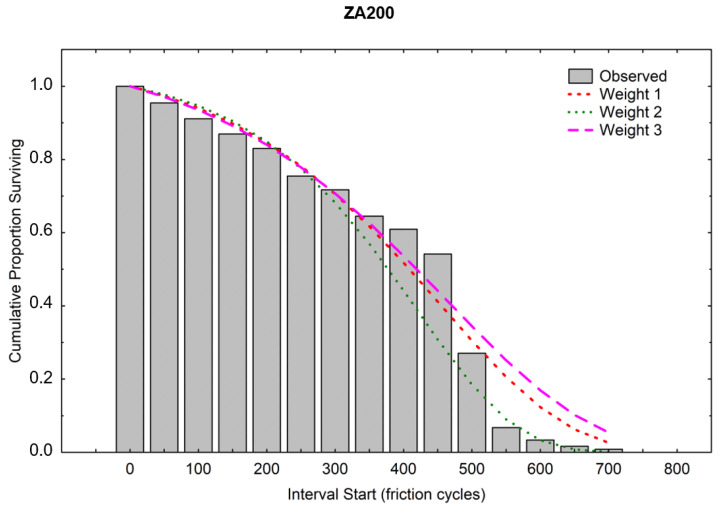
Cumulative proportion surviving (ZA200).

**Figure 14 materials-16-01779-f014:**
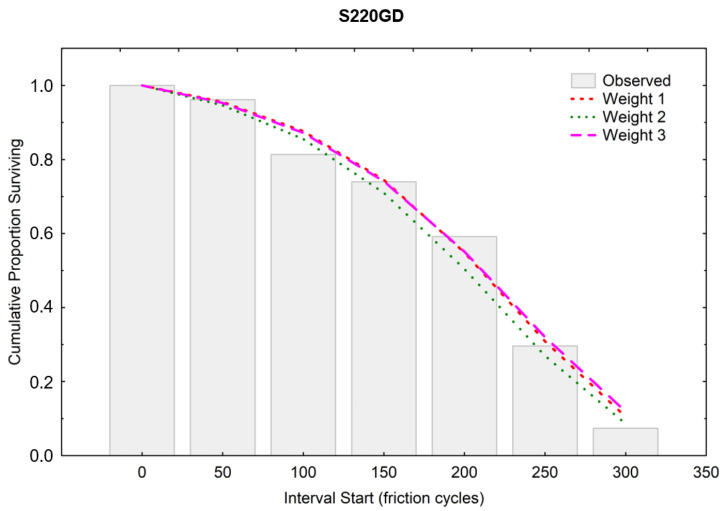
Cumulative proportion surviving (S220 GD).

**Figure 15 materials-16-01779-f015:**
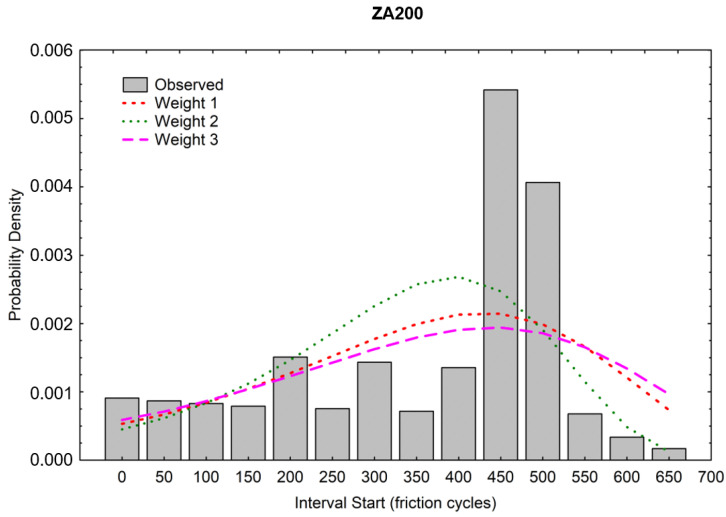
Probability density of the wear of the ZA200 coating.

**Figure 16 materials-16-01779-f016:**
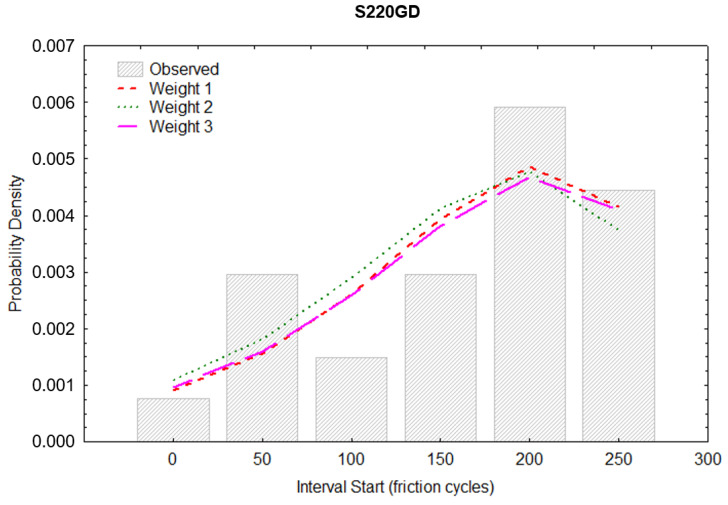
Probability density of the wear of the S220 coating.

**Figure 17 materials-16-01779-f017:**
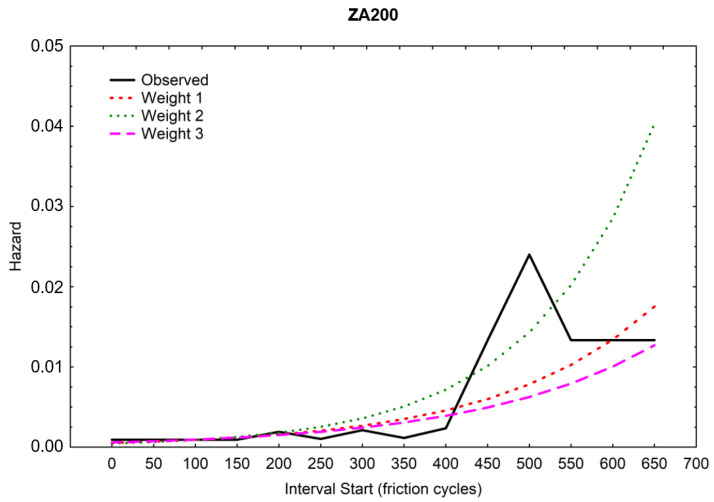
Hazard function for the ZA200 sheet metal (“black”).

**Figure 18 materials-16-01779-f018:**
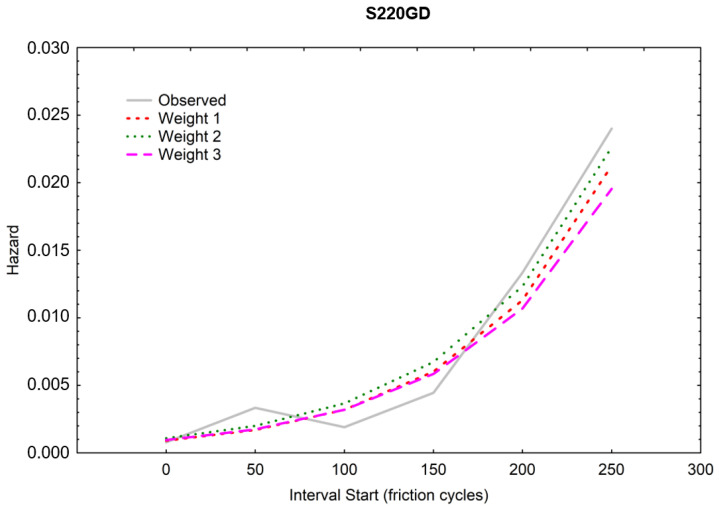
Hazard function for the S220GD sheet metal (“white”).

**Table 1 materials-16-01779-t001:** Selected physical properties of the research coatings.

Coating Designation	Minimum Total Coating Mass, Both Surfaces g/m^2^		Theoretical Guidance Values for Coating Thickness Per Surface in the Single-Spot Test, µm		Density g/cm^3^
	Triple-Spot Test	Single-Spot Test	Typical Value	Range	
Zinc–aluminum alloy coating masses (ZA)					
ZA095	95	80	7	5 to 12	6.6
ZA130	130	110	10	7 to 15	
ZA185	185	155	14	10 to 20	
ZA200	200	170	15	11 to 21	
ZA255	255	215	20	15 to 27	

**Table 2 materials-16-01779-t002:** Grubbs test results (sliding friction test results).

Coatings	Trimmed Mean—5.0000%	Winsorized Mean—5.0000%	Grubbs Test— Statistics	*p*
ZA200	465.56	459.45	2.20	0.23
S220GD	214.64	211.31	1.99	0.80

**Table 3 materials-16-01779-t003:** Results of the Mann–Whitney (M-W) *U*-test.

Rank Sum ZA200	Rank Sum S220GD	U	Z	*p*
204.0	96.0	5.0	3.852780	0.000117

## Data Availability

The data presented in this study are available on request from the corresponding author.

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
