# Peer review of "Research on the Durability and Reliability of Industrial Layered Coatings on Metal Substrate Due to Abrasive Wear"

_materials, 2023, doi:10.3390/ma16051779_

Round 1

Reviewer 1 Report

“Research on durability and reliability of industrial layered coatings on metal substrate due to abrasive wear" were studied by ball on disk performance test with only two coating types. Approaches were made with a lot of statistical analysis. The work presentation organization is not done very well. Detailed information about the novelty of the study is not highlighted. However, although it is not very interesting, it can be re-evaluated in the light of the following regulations.

1.       Why only two types of coatings were chosen and what is the difference and originality of this from other works. It should be clearly emphasized.

2.       Rewrite the abstract with the material, method and striking results.

3.       Please organize the experimental study and method part in a more concise and understandable way.

4.       Material and method are not presented very well. Table headers and table contents are not meaningful. Table 1 and Table 2 are given for what purpose. What does it mean to give so much standard information in Table 2?

5.       Figures 2 and 3 can be given with a single figure number. In Figure 2, the white coating can be marked with the thickness gauge.

6.       What is meant by the expression "..agglomerates of particles not seen" in line 126? An explanation needs to be added.

7.       In Figure 4 and 5, it would be more appropriate to give the coating names in the subtitles instead of just black and white.

8.       In all graphics, it is beneficial to write a little larger text on the scale. It's a little difficult to read.

9.       Strengthen and discuss your work with further metallurgical microstructural analysis. The adhesion of the coating to the substrate is mechanical or diffusion-based. A little detail should also be given about the properties of the base material.

Author Response

The response to reviews is attached in a separate file.

Reviewer 2 Report

The manuscript is about "Research on durability and reliability of industrial layered 2 coatings on metal substrate due to abrasive wear" the following comments should be considered before publication:

1. The chemical composition of the coating should be tabulated.

2. The method of coating application should be schematically shown.

3. The SEM image is not high resolution. Please specify the substrate and coating.

4.In present form the manuscript is an industrian report. To increase its scientific value please use some references and compare your results with them.

Author Response

(The authors gave the same response as above.)

Round 2

Reviewer 1 Report

Dear author, The requested corrections and additions have been made. In this state, it has become acceptable.

Reviewer 2 Report

The manuscript was improved significantly.